# A MILP Optimization Method for Building Seasonal Energy Storage: A Case Study for a Reversible Solid Oxide Cell and Hydrogen Storage System

**Oscar Lindholm [1],*** , **Robert Weiss [1]** , **Ala Hasan [1]** , **Frank Pettersson [2] and Jari Shemeikka [1]**

[1]  VTT Technical Research Centre of Finland, FI-02044 VTT Espoo, Finland; robert.weiss@vtt.fi (R.W.);
     ala.hasan@vtt.fi (A.H.); jari.shemeikka@vtt.fi (J.S.)

[2]  Laboratory of Process and Systems Technology, Faculty of Science and Engineering, Åbo Akademi
     University, Fi-20500 Turku, Finland; frank.pettersson@abo.fi

*  Correspondence: oscar.lindholm@vtt.fi

**Abstract:** A new method for the optimization of seasonal energy storage is presented and applied in a case study. The optimization method uses an interval halving approach to solve computationally demanding mixed integer linear programming (MILP) problems with both integer and non-integer operation variables (variables that vary from time step to time step in during energy storage system operation). The seasonal energy storage in the case study uses a reversible solid oxide cell (RSOC) to convert electricity generated by solar photovoltaic (PV) panels into hydrogen gas and to convert hydrogen gas back to electricity while also generating some heat. Both the case study results and the optimization method accuracy are examined and discussed in the paper. In the case study, the operation of the RSOC and hydrogen storage system is compared with the operation of a reference system without energy storage. The results of the study show that installing an RSOC and hydrogen storage system could increase the utilization of onsite renewable energy generation significantly. Overall, the optimization method presents a relatively accurate solution to the case study optimization problem and a sensibility analysis shows a clear and logical pattern.

**Keywords:** energy storage; hydrogen; power-to-gas; reversible solid oxide cell; optimization; mixed integer linear programming; MILP

## 1. Introduction

During the last decade, people have become increasingly more aware of the environmental impact of their daily routines and consumption behavior. Minimizing the carbon dioxide footprint by replacing fossil fuels with renewable energy sources is therefore a change process driven by the end consumers of products and services. Meanwhile, in the building sector, there has been a trend towards "greener" solutions and more energy efficient constructions. Concepts like net zero energy buildings (NZEB) and nearly zero energy buildings (nZEB) have been studied in several research projects [1,2]. These concepts are striving towards covering a higher share of the building energy demand with on-site generated renewable energy.

Seasonal energy storage can be used to increase the on-site utilization of solar energy installations, like photovoltaic (PV) panels and solar thermal collectors. Storing energy during periods of high on-site energy generation and utilizing the stored energy during periods of low on-site energy generation will increase the utilization rate of the solar energy installations for individual buildings. Hence, seasonal energy storage is an important contributing factor for buildings that are targeting a net zero annual energy balance.

Investing in energy storage also has an economic driving factor. Energy import prices are namely often significantly higher than export prices for household energy consumers [3–5]. By storing energy, household consumers can avoid selling cheap excess energy in the summer and buying it back with a significantly higher price in the winter. The economic benefits of seasonal energy storages are additionally boosted by the falling installation costs of solar PV technologies. Between 2010 and 2018, the total installation cost of solar PV panels dropped by 74% [6].

Optimization-based design methods can be used to maximize the utilization and minimize the cost and environmental impacts of seasonal storage systems. These problems are usually formulated as time series optimization problems, where the design variables set constraints on the system operation in every time step of the annual operation. Different kinds of stochastic optimization method have been used for this purpose. Durão et al. (2014) studied the advantages of using genetic algorithms to optimize a seasonal energy storage of solar thermal energy [7]. Other approaches, such as particle swarm optimization [8] and simulated annealing (SA) [9], have also been used for optimizing similar energy storage systems. Zhang et al. (2018) used harmony search and chaotic search methods based on a SA approach to optimize renewable energy systems including different energy storages [10]. The objective of the optimization by Zhang et al. was to minimize the life-cycle cost of the renewable energy system.

Mixed integer linear programming (MILP) is a demanding optimization problem category. MILP problems include both integer and real variables and tend to become computationally expensive as the number of integer variables increases. Kotzur et al. (2018) proposed a clustering method to solve MILP problems related to the optimization-based design of energy storage systems [11]. The integer variables in the method by Kotzur et al. are of a binary nature and define whether certain components in an energy system exist or not. Steen et al. (2014) solved similar MILP energy storage problems to minimize both the cost and the emissions of a thermal energy storage system by optimizing the system operation and setup [12]. Moreover, Wang et al. (2015) proposed a MILP-based control method that uses day-ahead pricing, weather forecasts and customer preferences to minimize the energy expenditures of a building energy system comprised of a battery and building-integrated solar PV [13]. Pinzon at al. (2017) propose a similar MILP-based control method for smart buildings with integrated solar PV and batteries [14]. Both of these control methods showed favorable results when testing them against simulation software and measured data, respectively.

This paper aims to present and evaluate a novel seasonal energy storage optimization method that uses a time interval halving approach to solve computationally expensive MILP problems. The method is unique since it accepts integer variables in every time step of the annual operation. This implies that the method can handle multi-mode devices. The use of integer control variables for multi-mode devices is essential when developing accurate operational models that are able to separate between different operational modes. The authors have not found any other optimization methods in the literature that can solve seasonal storage optimization problems that include this type of integer control variable.

A case study is presented in this paper in order to examine and evaluate the presented method. The case study examines the optimal operation and design of a seasonal energy storage system for an office building with an over-production of solar energy during the summer season. The seasonal energy storage system uses reversible solid oxide cell (RSOC) technology to convert electrical energy generated by PV to hydrogen gas and to convert hydrogen gas back to electricity, while also generating some heat. The case study is based on a preceding study presented in [15], where the energy storage was only used for short-term grid balancing and did not yet cover the seasonal aspects.

Earlier studies on hydrogen storage system optimization have been done by Castañeda et al. (2013), Luta and Raji (2018), as well as Carapellucci and Giordano (2011) [16–18]. The hydrogen storage systems in these studies use separate electrolyzers and fuel cells and are thus not comparable with the RSOC and hydrogen storage system used in the case study in this paper.

## 2. Methods

### 2.1. Optimization Method

The novelty of the optimization method is that it can solve computationally expensive seasonal storage operation optimization problems with both integer and non-integer operation variables (variables that vary from time step to time step during operation). This means that the method can optimize energy storage systems containing multi-mode devices (e.g., on-off mode devices and reversible fuel cells) where integer variables are used to control operation modes. This type of optimization problem tends to become too computationally expensive for conventional optimization as the number of integer variables increases.

The basic idea of the method is to first perform a rough optimization of the annual operation, and then gradually fine-tune the solution by re-optimizing smaller time intervals. The first step of the method is, hence, to optimize the annual operation using a relatively small number of time steps, $N$, ($N = 26$ time steps are used in the case study). Thereafter, the energy storage load at the beginning, the middle and the end of the one-year period is fixed. The one-year period is then bisected, so that the first half of the year forms one new time period and the second half of the year forms another new time period.

In the second optimization step, the new time intervals are re-optimized separately, using the same number of time steps ($N$) as in the first optimization step, and the storage loads at their center points (not the starting points and the end points) are fixed. Then, the time intervals are bisected, and new intervals are formed. The second optimization step is repeated until the time steps reach the desired size (seven hours in the case study). The time step will reduce in size for every new interval bisection since the number of time steps remains the same, while the time interval size shrinks.

A graphical explanation of the first and second optimization steps are presented in Figure 1. In the figure, $m_{i,j}$ is the energy storage load at the central point of time interval j in optimization step i. Step 1 also shows the storage load at the beginning, a, and the end, b, of the year. To guarantee a net zero annual storage balance, the storage loads at point a and b must be equal to one another. Otherwise, the energy storage cannot operate in a sustainable way for several years.

After each optimization step, the new center points of the new time intervals will be added to the final solution of the annual system operation. This means that the storage load for all the points that are fixed will remain constant throughout the optimization process. Hence, the storage load for $2^{(n-1)}$ (where n is the number of the optimization step) new points will be added to the final solution from each optimization step, except for the first optimization step, where three points (center point, starting point and end point) are added to the final solution.

Each time interval optimization is solved with a Matlab R2019b solver called intlinprog, which uses a branch-and-bound approach to tackle MILP problems [19]. The solver is also able to identify and eliminate futile subproblem candidates by adjusting the constraints of the optimization problem.

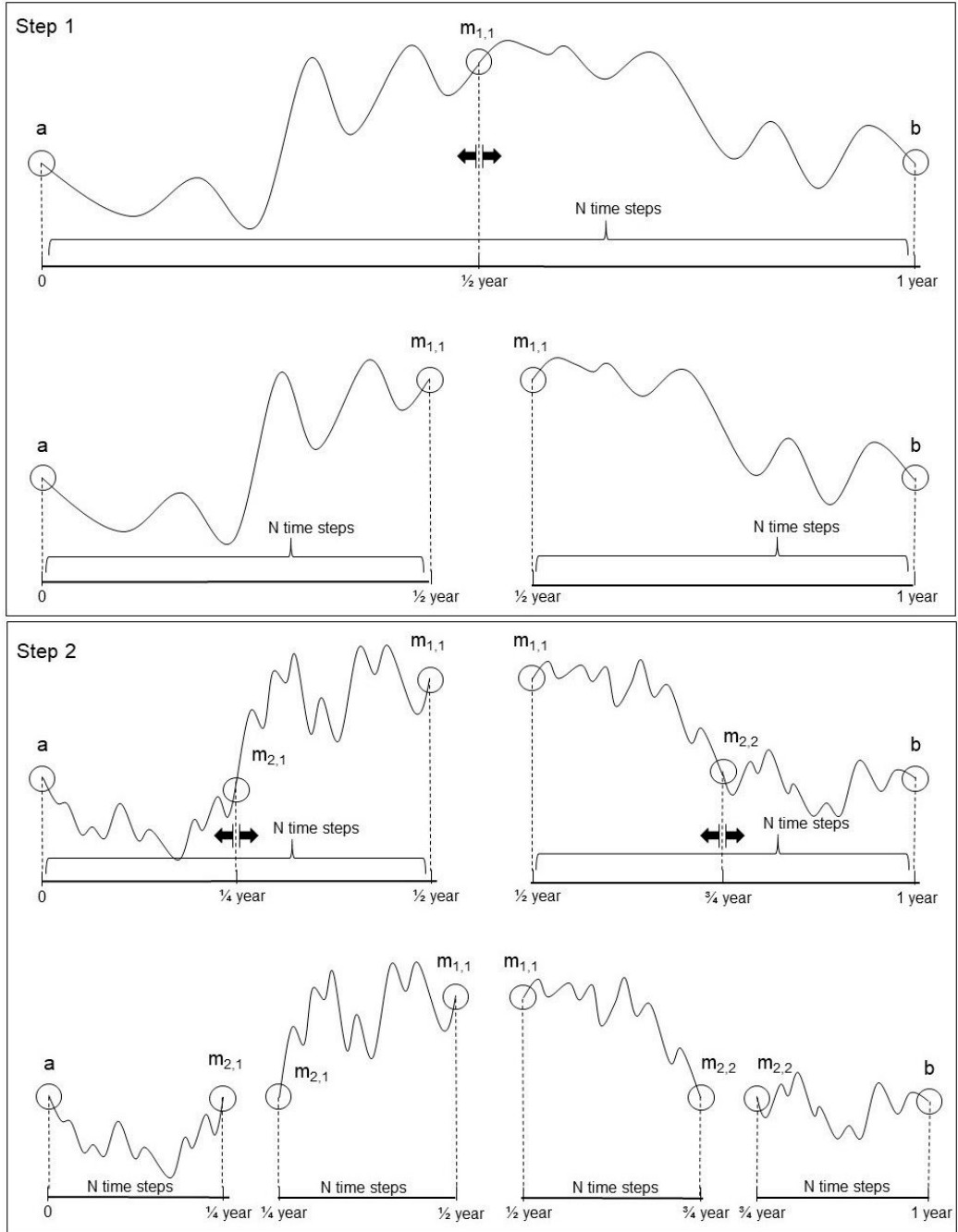

**Figure 1.** Graphical description of the first and second optimization step in the interval halving optimization method.

*2.2. Case Study: Reversible Solid Oxide Cell and Hydrogen Storage System for Seasonal Storage of Solar Energy*

A reversible solid oxide cell (RSOC) is an electrochemical device that can operate either as a solid oxide electrolysis cell (SOEC) or as a solid oxide fuel cell (SOFC). In SOEC mode, the RSOC consumes electricity and water in order to produce hydrogen and oxygen through a redox reaction. In SOFC mode, the redox process is run in reverse, so that the RSOC consumes hydrogen and produces electricity, heat and water [20,21]. An RSOC together with a hydrogen compressor and a compressed hydrogen storage can be used as an energy storage system which is able to store energy in the form of hydrogen gas. Such a system is hereinafter referred to as an RSOC and hydrogen storage (RSOCHS) system. The power input capacity of the RSOC in SOEC mode is significantly higher than the power

output capacity in SOFC mode [22]. This means that discharging the RSOCHS storage is more time consuming than charging the storage.

In the context of the case study, the RSOCHS system is used in an energy system to balance out large seasonal variations in solar PV generation for a 7874 m$^2$ floor-area office building, VTT FutureHub, (Figure 2), located in Espoo, Finland. The building characteristics are presented in Table 1. The location of the building is 60°11′11.4″ N 24°48′49.0″ E and the Köppen climate classification of the site climate is Dfb [23].

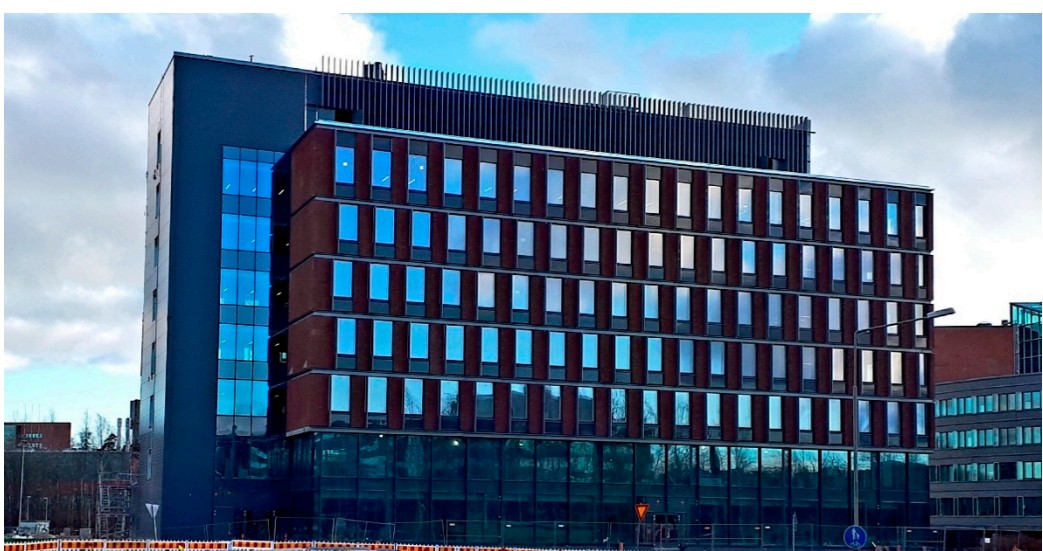

**Figure 2.** The office building, VTT FutureHub, in Espoo, Finland.

**Table 1.** VTT FutureHub building characteristics.

| Floor Area | 7874 m$^2$ |
|---|---|
| Volume | 29,483 m$^3$ |
| Roof area | 1163 m$^2$ |
| Construction year | 2020 |

The assumed energy system of the office building (depicted in Figure 3) includes solar PV panels, an RSOCHS system, a water tank, a boiler with a hydrogen combustor for heat generation as well as connections to the electricity grid and the district heating network (for both import and export). The annual heating and electricity demand profiles of the office building are produced by IDA ICE simulations [24]. These profiles are presented in Figure 4.

The optimization method presented in this paper is used to optimize the system operation, where the objective function is to minimize the annual operating expense (OPEX). Due to the differences in the two RSOC operation modes, they must be modelled as separate functions in the optimization problem. A binary decision variable, which determines the operation mode, must also be introduced at each time step in the optimization problem. The other variables in the optimization problem consist of energy transfer rates, which can be described as non-integer variables. The optimization problem is considered as a computationally expensive MILP problem due to its high number of integer and non-integer variables. The problem characteristics are, hence, ideal for the interval halving optimization method presented in this paper. Table 2 presents the dimensions of the case study optimization problem.

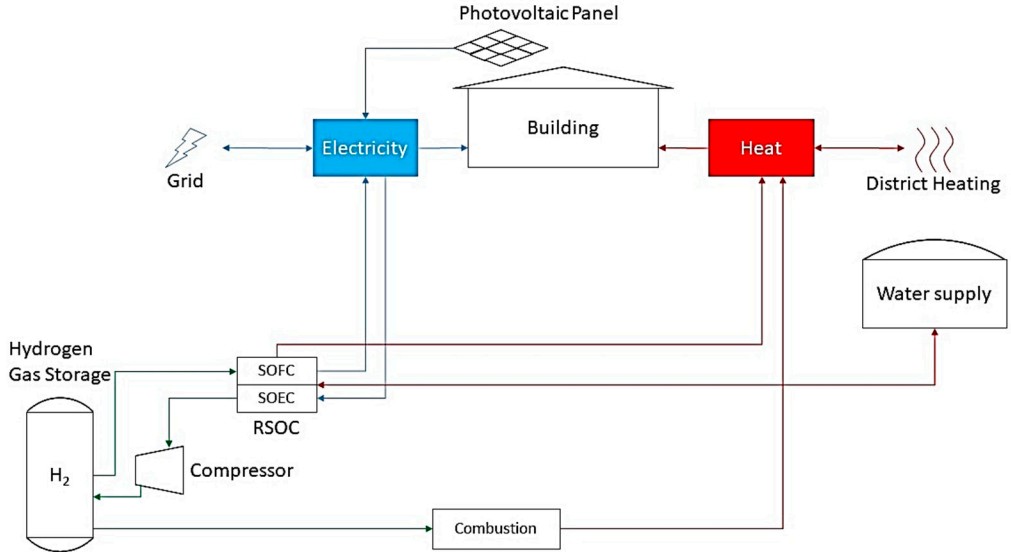

**Figure 3.** The reversible solid oxide cell and hydrogen storage system (RSOCHS).

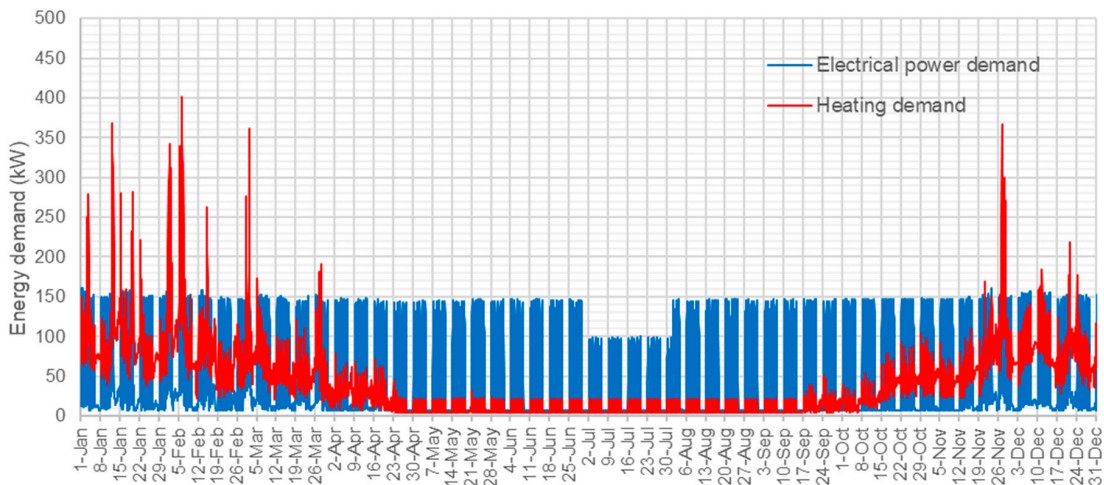

**Figure 4.** Electricity and heating demand profiles of the office building.

**Table 2.** Statistics of the optimization problem as well as the central processing unit (CPU) time required to solve the problem.

| **Number of Binary Variables** | **1248** |
| --- | --- |
| Number of non-integer variables | 9985 |
| Number of constraints | 6240 |
| CPU time | 15 s |

### 2.2.1. Calculated Cases

The optimal operation of three RSOCHS systems with different RSOC sizes is generated for different solar PV areas and hydrogen storage capacities. The RSOC sizes used for the RSOCHS systems are 20/80 kW (20 kW maximum power output in SOFC mode and 80 kW maximum power input in SOEC mode), 50/200 kW and 100/400 kW. All tested case parameters are presented in Table 3.

**Table 3.** Tested case parameters.

| Case Parameter | Tested Values |
| --- | --- |
| RSOC sizes (kW) | 20/80, 50/200, 100/400 |
| Solar PV areas (m$^2$) | 500, 1000, 2000, 3000, 4000, 5000, 6000, 7000, 8000, 9000, 10.000 |
| Hydrogen storage (kWh) | 10, 20, ... , 600 |

The RSOCHS systems are compared with a reference system without any energy storage in order to identify the strengths and weaknesses of the RSOCHS system. By comparing the energy systems, it is also possible to examine whether the RSOCHS system is a suitable solution for seasonal storage or not.

### 2.2.2. Objective Function

The objective of the optimization problem is to minimize the annual OPEX of the RSOCHS system by using the operating mode and the different energy transfer rates as variables. The OPEX of the system in this context is defined as the sum of the electricity and district heating costs minus the income generated by energy export to the grid and the district heating network. The objective function is, thus, expressed as follows:

$$\min \sum_{i=1}^{N} (\dot{Q}_{im,i} C_{Qim,i} + P_{im,i} C_{Pim,i} - \dot{Q}_{ex,i} C_{Qex,i} - P_{ex,i} C_{Pex,i})t, \tag{1}$$

where $C_{Qim,i}$ is the district heating import price at time step $i$, and $C_{Pim,i}$ is the electricity import price at time step $i$, which is sum of the electricity spot price, the electricity distribution tariff and the grid tax. $C_{Qex,i}$ and $C_{Pex,i}$ are the export prices for heat and electricity at time step $i$. The variables $\dot{Q}_{im,i}$ and $\dot{Q}_{ex,i}$ are the imported and exported heat rate at time step $i$, while the variables $P_{im,i}$ and $P_{ex,i}$ are the imported and exported electrical power at step $i$, respectively. The time step size, $t$, and the number of time steps, $N$, are selected so that the sum of all time steps is equal to one year.

### 2.2.3. RSOC Functions

In SOEC mode, the RSOC is modelled with one single function that describes the hydrogen output as a linear function of the electrical power input, $P_{E,in,i}$, and the binary decision variable, $\delta_i$, for each time step $i$. The SOEC function is expressed as:

$$f_E(P_{E,in,i}, \delta_i) = a_E \delta_i + k_E P_{E,in,i}, \tag{2}$$

where the coefficients $a_E$ and $k_E$ are dependent on the size and operating range of the RSOC device.

In SOFC mode, two different linear functions are used; one for the hydrogen input, $f_F$, and one function for the thermal energy output, $f_G$. Both of these functions are dependent on the electrical power output of the RSOC. These functions are expressed as follows:

$$f_F(P_{F,out,i}, \delta_i) = a_F \delta_i + k_F P_{F,out,i}, \tag{3}$$

$$f_G(P_{F,out,i}, \delta_i) = a_G \delta_i + k_G P_{F,out,i}. \tag{4}$$

The coefficients $a_F$, $a_G$, $k_F$ and $k_G$ are here also dependent on the size and operating range of the RSOC device.

### 2.2.4. Constraints

There are five constraints for each time step in the optimization problem. These constraints consist of two energy balances, one for electrical energy and one for thermal energy, two constraints for the capacity of the RSOC and one constraint for the hydrogen storage.

The electrical energy balance is the sum of all electrical energy transfer rates and is expressed as:

$$P_{F,out,i} - P_{E,in,i} + P_{im,i} - P_{ex,i} - P_{d,i} + P_{PV,i} - \frac{h_{c,out} - h_{c,in}}{\eta_c \text{LHV}} f_E(\delta_i, P_{E,in}) = 0 \; \forall i \tag{5}$$

where the parameters $P_{d,i}$ and $P_{PV,i}$ are the electrical power demand of the building and the power generated by the solar PV panels at time step $i$. The last part of the equation describes the power used by the hydrogen compressor at time step $i$, where $\eta_c$ is the total efficiency of the hydrogen compressor, LHV is the lower heating value of hydrogen gas. $h_{c,in}$ and $h_{c,out}$ are the specific enthalpies of the hydrogen gas at the inlet and outlet of the hydrogen compressor.

The thermal energy balance is expressed as follows:

$$f_G(\delta_i, P_{F,out,i}) + \dot{Q}_{b,i} + \dot{Q}_{im,i} - \dot{Q}_{ex,i} - \dot{Q}_{d,i} = 0 \; \forall i \tag{6}$$

where $\dot{Q}_{b,i}$ is the heat generated through combustion of hydrogen, and $\dot{Q}_{d,i}$ is the heating demand of the building at time step $i$.

The input, $P_{E,in,i}$, and output, $P_{F,out,i}$, power ranges of the RSOC device are dependent on the size of the RSOC device as well as the operating mode. The RSOC power input and output constraints are thus controlled by the decision variables for each time step. These constraints are expressed as:

$$P_{E,in,max}\delta_i \geq P_{E,in,i} \geq P_{E,in,min}\delta_i \; \forall i \tag{7}$$

$$P_{F,out,max}(1 - \delta_i) \geq P_{F,out,i} \geq P_{F,out,min}(1 - \delta_i) \; \forall i \tag{8}$$

The hydrogen constraint is dependent on the storage load at the beginning of each time step, which is the cumulative sum of the produced hydrogen plus the initial stored hydrogen, $E_{H,0}$, minus the cumulative sum of consumed hydrogen. The hydrogen storage constraint is thus dependent on the system operation of earlier time steps, which is the reason why each time step cannot be optimized individually. Hydrogen storage constraints are expressed as:

$$E_{H,cap} \geq E_{H,0} + t\sum_{s=1}^{i}\left(f_E(P_{E,in,s}, \delta_i) - f_F(P_{F,out,s}, \delta_i) - Q_{b,s}\right) \geq 0 \; \forall \; i \in \{1 \ldots (N-1)\} \tag{9}$$

$$E_{H,cap} \geq E_{H,0} + t\sum_{s=1}^{N}\left(f_E(P_{E,in,s}, \delta_i) - f_F(P_{F,out,s}, \delta_i) - Q_{b,s}\right) \geq E_{H,0} \tag{10}$$

where the parameter $E_{H,cap}$ is the maximum capacity of the hydrogen storage. The constraint for the last time step in Equation (10) is different from the constraint for the other time steps in Equation (9) since it only accepts a final storage load above the initial stored energy.

### 2.2.5. Assumptions

The parameters in the case study are comprised of the annual energy demand, the solar PV generation and the energy price profiles as well as some technical data regarding the RSOC, the hydrogen compressor and the solar PV panels. The RSOC performance data are based on confidential data provided by the RSOC development team at the Technical Research Centre of Finland and are therefore not presented in this paper.

According to the US Department of Energy, the isentropic efficiency, $\eta_{is}$, of hydrogen compressors used for small hydrogen gas terminals is about 65% [25]. The electrical motor efficiency, $\eta_m$, of the compressor is assumed to be 95%. The total efficiency of the hydrogen compressor is thus assumed to be:

$$\eta_c = \eta_{is}\eta_m \approx 62\% \tag{11}$$

The solar PV production capacity is assumed to be 0.17 kWp/m$^2$ and the PV generation profile is based on simulated data for a system where 50% of the solar panels are facing east and 50% are facing west [26].

As a power and heat price scenario base, we use current prices and tariffs for Espoo, Finland. As mentioned earlier, the electricity import price is the sum of the electricity spot price, the distribution tariff and the grid tax. The electricity spot price used in this study is based on the Nord Pool spot price data from 2017 (Figure 5) [27] and the distribution tariff is selected according to the pricing of Caruna Oy, which is the electricity distributor in Espoo [28]. The grid tax in Finland is 2.253 c/kWh for non-industrial consumers [29]. The electricity export price is based on the price the Finnish energy company Fortum Oy pays for excess electricity generated by households [3]. This price is the hourly spot price minus a 0.24 c/kWh transfer fee. The average export and import electricity prices are summarized in Table 4.

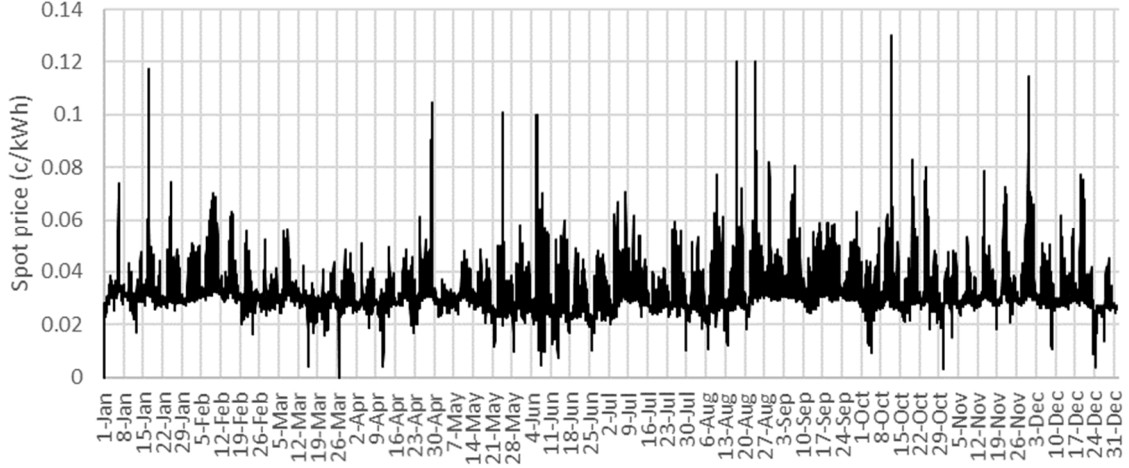

**Figure 5.** Nord Pool electricity spot prices 2017 [27].

**Table 4.** Electricity price and tariffs for the case building in Espoo, Finland.

|  | Export (c/kWh) | Import (c/kWh) |
|---|---|---|
| Electric energy | Nord Pool spot price Average: 3.30 | Nord Pool spot price Average: 3.30 |
| Daytime distribution tariff, winter [1,2] | - | 2.42 |
| Other time distribution tariff [1] | - | 1.15 |
| Transfer fee | −0.24 | - |
| Grid tax | - | 2.25 |
| **Total average price** | **3.06** | **9.12** |

[1] Electricity distribution tariff for a 400 V connection; [2] Daytime distribution tariff, winter: Mon–Sat 7am–10pm, Nov–Mar.

The district heating import and export prices presented in Figure 6 are also based on pricing by Fortum Oy, who operates the district heating network in Espoo [4]. These prices vary from month to month, and are usually much higher during the winter months, when the heating demand is higher. The export prices are dependent on the temperature of the provided heat, but since temperatures are

not considered in the optimization model, the export prices in Figure 6 are based on the average export price for each month.

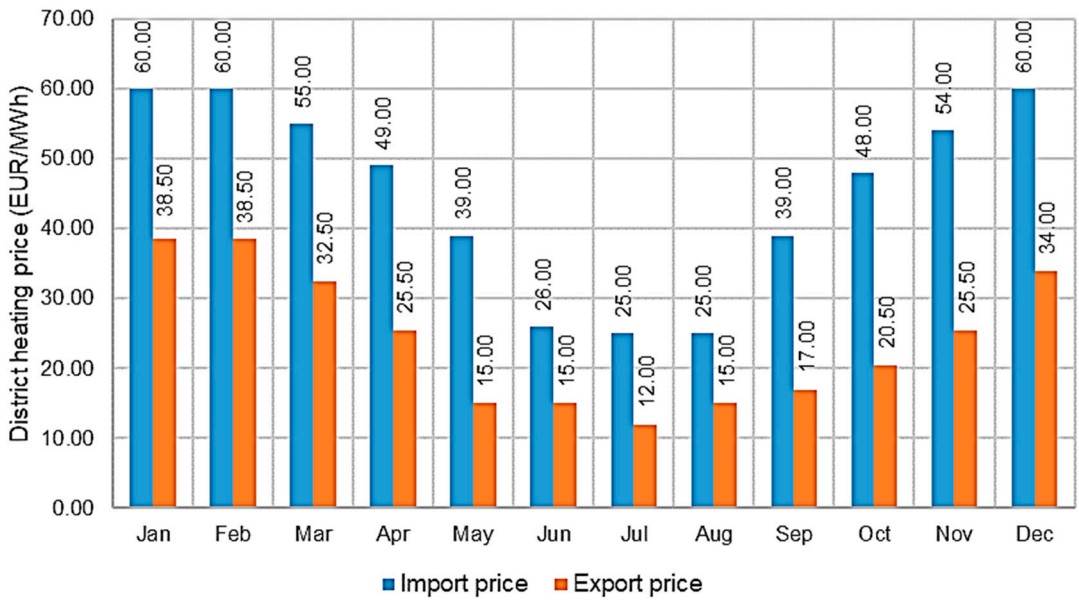

**Figure 6.** District heating import and export prices in Espoo, Finland [4,5].

## 3. Results

### 3.1. Results of the Case Study

Three RSOCHS systems as well as one reference system were optimized for different solar PV generation capacities using the interval halving optimization method presented in the paper. The reference system has no onsite energy generation or storage. It is only comprised of connections to the electricity grid, the district heating network and the solar PV panels. The idea of the optimization was to optimize the operation of the system by minimizing the OPEX. The results of the optimization are presented and analyzed in this chapter in order to form a conception of the behavior and benefits of a RSOCHS system as a seasonal energy storage for buildings.

The optimal operation of the different RSOCHS systems was analyzed for three different solar PV installation areas (1000, 2000, and 3000 m²), with results presented in Figures A1–A9 in the Appendix A. All of the systems show the same trend; storing energy during summer and consuming the stored energy during winter. One surprising remark, which also holds true for all storage systems, is that the system uses large amounts of electricity from the grid to fill up the hydrogen storage, even during the summer, when there is a surplus of energy generated by the solar PV panels. This indicates that it is cost-effective to convert electricity from the grid to hydrogen gas and then use it for heating during the winter.

By comparing the optimal operation of different system setups, it is shown that seasonal energy storage has some benefits for energy systems with large solar PV installations. This remark is supported by the results presented in Figures 7–10. It shall, however, be pointed out that the optimization model is formulated so that it does not allow the RSOC device to be turned off, since it might be complicated to ramp the RSOC up and down due to the high operation temperature. This is the reason why the RSOCHS systems perform worse than the reference system for smaller PV installations.

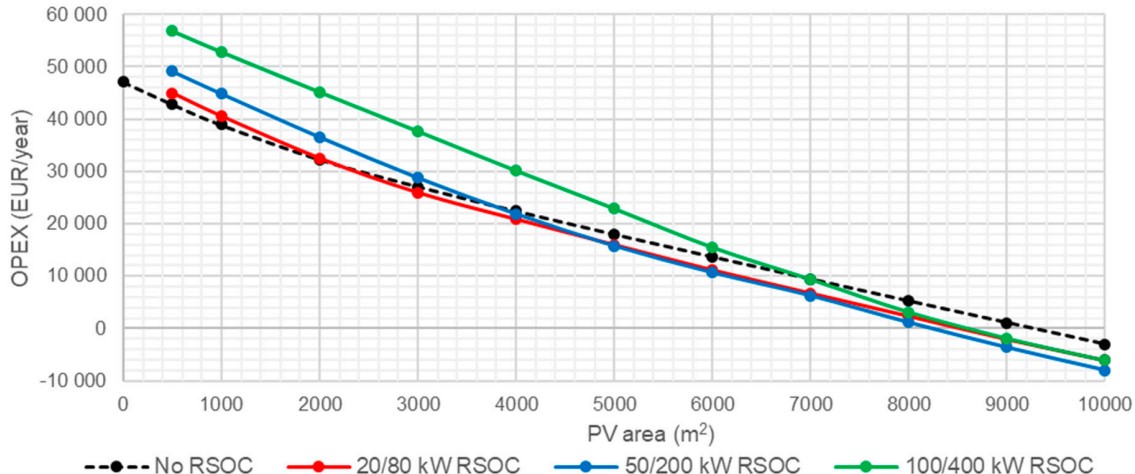

**Figure 7.** Optimal annual operating expense (OPEX) (for the optimal hydrogen storage size) as a function of installed solar photovoltaic (PV) panel area for the three RSOCHS systems and the reference system.

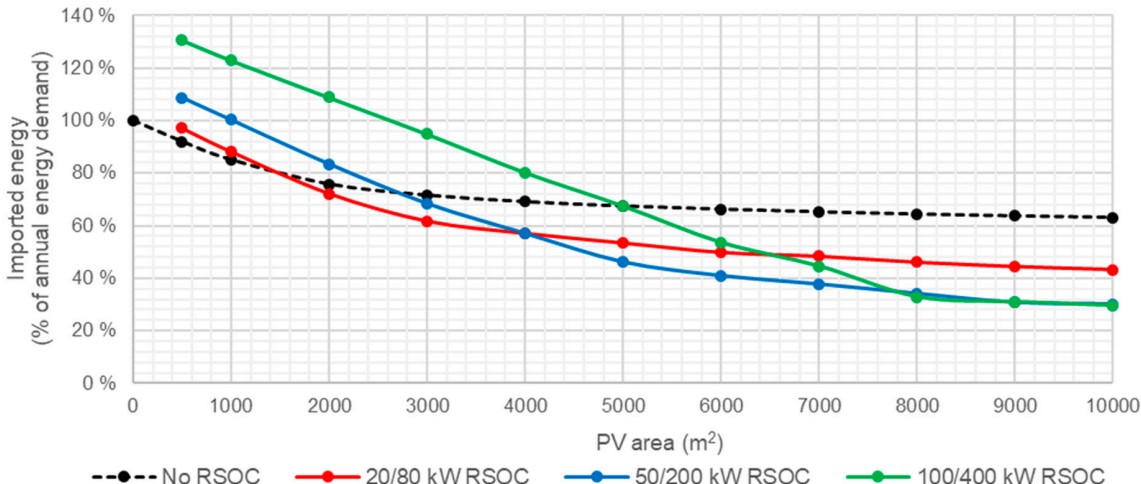

**Figure 8.** Annual imported energy (electricity and heat) (for the optimal hydrogen storage size) as a function of installed solar PV panel area for the three RSOCHS systems and the reference system.

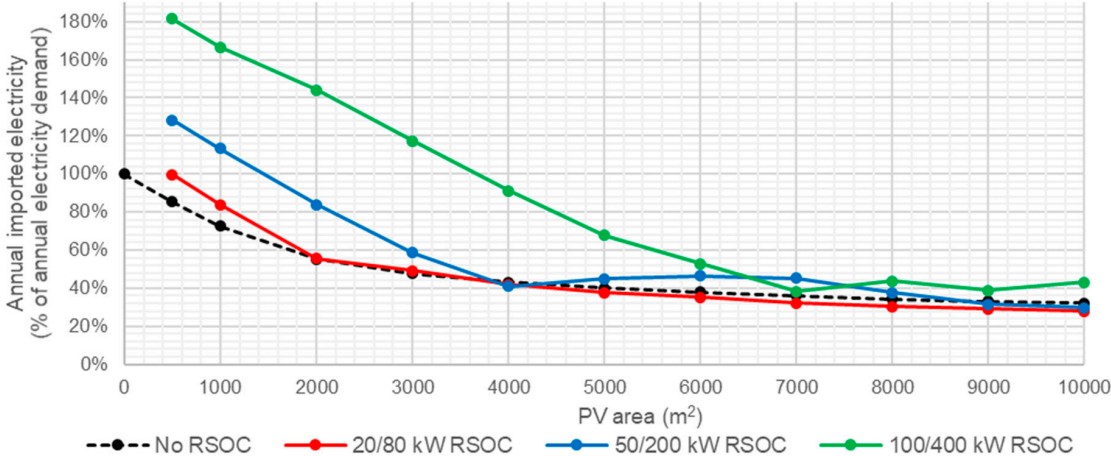

**Figure 9.** Annual imported electricity (for the optimal hydrogen storage size) as a function of installed solar PV panel area for the three RSOCHS systems and the reference system.

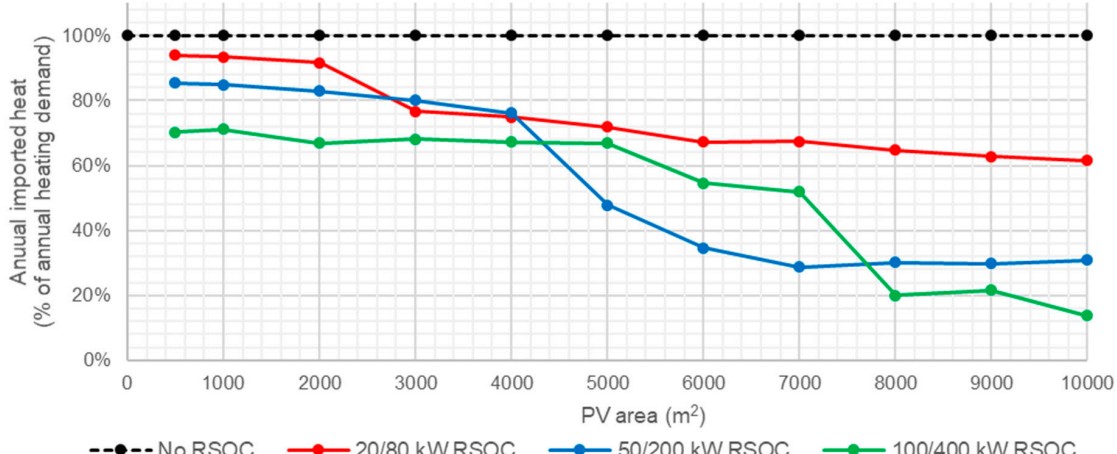

**Figure 10.** Annual imported heat (for the optimal hydrogen storage size) as a function of installed solar PV panel area for the three RSOCHS systems and the reference system.

Figure 7 shows the optimal OPEX (for the optimal hydrogen storage size) as a function of installed PV area for the RSOCHS storages systems and the reference energy system (no RSOC). By examining the graph in Figure 7, it can be observed that increasing the solar PV area has a higher cost saving impact on the RSOCHS system than on the reference system. More installed PV panels entail a lower OPEX and an increased advantage of the RSOCHS system. It can also be observed that the optimal size of the RSOC is dependent on the installed PV area, as small PV areas appear to be more suitable for small RSOC devices, while the power generation of larger PV areas can be utilized to a greater extent by a big RSOC device.

The same trend can be observed when the amount of imported energy is examined for the different energy systems. Figure 8 shows that the amount of imported energy can be reduced by investing in an RSOC and hydrogen storage system. It can also be observed that the energy savings increase when more solar PV panels are added to the system. This proves that the RSOCHS system works as intended, halving the need for annually imported energy for energy systems with large solar PV installations.

Figures 9 and 10 show that reduced district heating consumption accounts for most of the energy savings generated by the RSOC and hydrogen storage system. The figures describing the optimal RSOC operation in the Appendix A show that most of the time, heat is generated by the RSOC, but the peaks in the heat generation are caused by hydrogen combustion. Some of these peaks may be a consequence of the interval halving approach in the optimization method.

Without hydrogen storage, the operation of the RSOCHS systems and the reference system would be the same. Hence, the difference in OPEX between the systems in Figure 7 is only induced by the hydrogen storage size. It can be noted that the impact of the hydrogen storage size on the OPEX is marginal for PV installations below 5000 m$^2$. For big PV installations, however, the significance of the hydrogen storage size is more crucial, especially when the OPEX of the systen drops below zero.

Figure 11 shows the optimal hydrogen storage capacity for different RSOCHS systems and solar PV areas. The figure indicates that the optimal size of the hydrogen storage increases with the size of both the solar PV installation and the RSOC device. The energy content of the hydrogen stroage is calculated using the LHV of hydrogen gas.

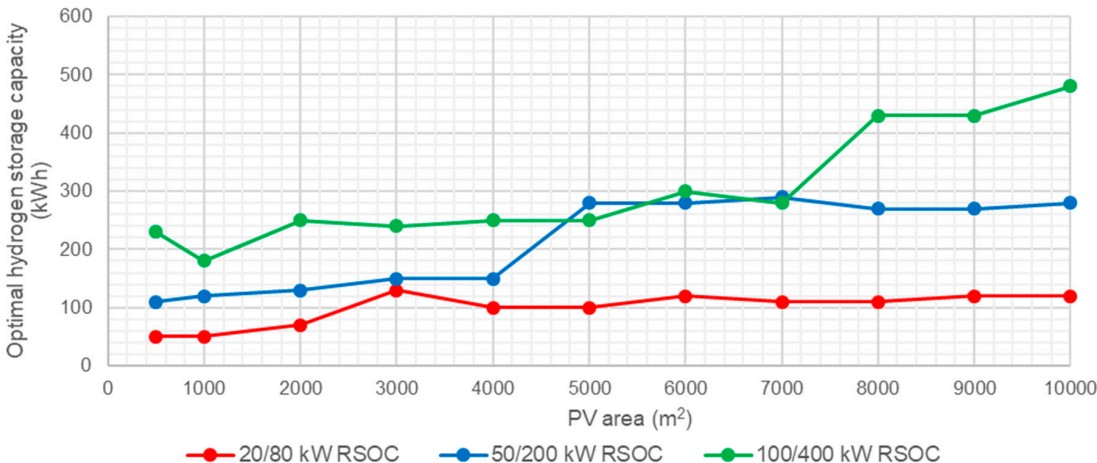

**Figure 11.** Optimal hydrogen storage size as a function of installed solar PV panel area for the three RSOCHS systems and the reference system.

### 3.2. Optimization Model Perfomance

The model is evaluated by examining the optimization results of various RSOCHS system setups and investigating whether an increased degree of freedom in the energy system can result in lower OPEX values or not. Figure 12 shows the optimal OPEX of the 50/200 kW RSOC system as a function of the hydrogen storage capacity for different solar PV installations. The OPEX should decrease with increasing storage capacity, since more storage entails more relaxed constraints in the optimization problem. This is however not exactly valid for all the results. In the graphs, it can be observed that the trend lines are all decreasing when the storage capacity is increasing, but this is not exactly followed by the individual optimized points. It can be presumed that this small but noticeable inconsistency in the results is caused by some inaccuracy of the optimization model.

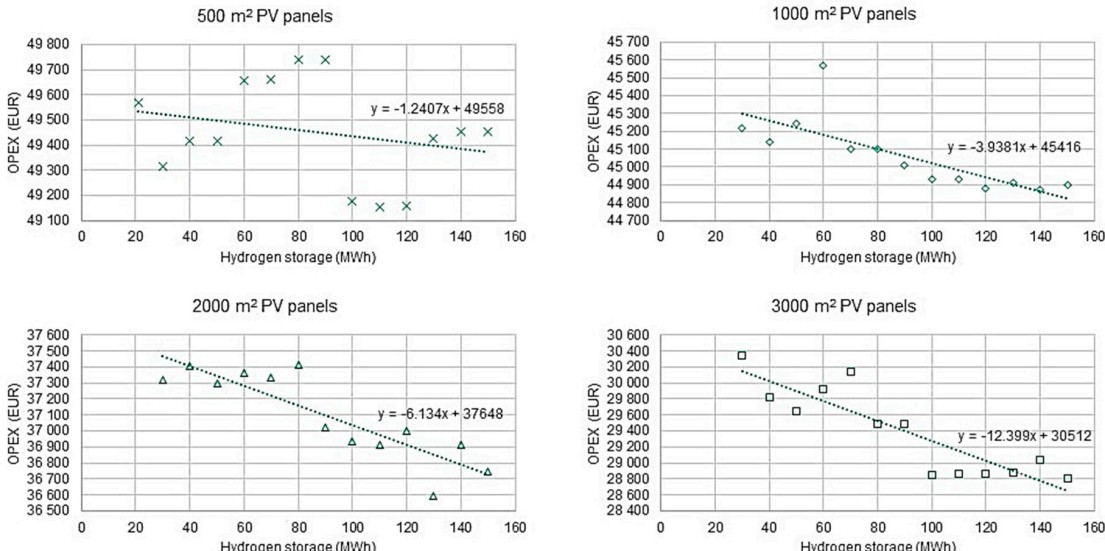

**Figure 12.** Optimal OPEX of the 50/200 kW RSOC system as a function of the hydrogen storage capacity for different solar PV installations.

One of the main sources of inaccuracy in the method could be the halving approach. Fixing a few points in the first optimizations steps will add more constraints on the following optimization steps which, in turn, will prevent the solution from reaching the global optimum. The peaks in the heat generation graphs in Figures A1–A9 in the Appendix A are visible indications of the inaccuracy

caused by the interval halving method. These peaks are most likely a consequence of forced hydrogen combustion due to the additional constraints produced by the interval halving process.

By analyzing deviation from the trend line in the graphs in Figure 12, it can be concluded that the optimization method is still relatively precise. The deviation from the trend line is namely below 2% for each optimized point in Figure 12. We also tried shifting the start of the year by 1400 h (58 days 8 h), but this action did only affect the value of the objective function (the OPEX) by less than 1%. This remark also supports the statement that the optimization method is relatively precise.

An additional drawback of the proposed optimization method is that the system component dimensions cannot be used as design variables in the optimization problem, as the optimization problem is divided into several subproblems with problem-speci0fic variables. Moreover, introducing the RSOC size as a variable would render the problem an MINLP (mixed integer non-linear programming) problem, which cannot be solved by the presented method. Hence, the optimal design of the energy storage system is found by performing several optimizations using different system component dimension combinations. This makes it complicated to use the method for the optimization-based design of energy storage systems with many different components, as it is impossible to include the component investment costs in the objective function.

A more accurate evaluation of the optimization method performance would require a comparative analysis with other optimization methods. For the time being, no other optimization tools are available for solving the type of optimization problems presented in the case study.

## 4. Conclusions

An interval halving MILP optimization method for seasonal storages was presented and tested by applying it to a case study. The novelty of the method is that it allows integer variables in each time step of the annual energy storage operation. The aim of the case study was to examine the optimal operation and setup of an RSOCHS system and evaluate its suitability as a seasonal energy storage. The operating principle of the RSOCHS system is to use RSOC technology to convert electrical energy to hydrogen gas and to convert hydrogen gas back to electricity while also generating some heat [15].

By analyzing the results of the optimization problem for the investigated case study, it can be noted that the solution of the optimization problem is a relatively accurate estimation of the optimal seasonal storage operation. Moreover, it shall be pointed out that the optimization problem presented in the case study is computationally too expensive to be solved by conventional optimization methods and that no other alternative optimization method suitable for the problem has been found in the literature so far. The model is hence ideal for optimizing MILP seasonal storage problems, but it could also be implemented in other computationally demanding linear scheduling optimization problems with cumulative constraints.

There are, however, a few minor shortcomings that can be discussed regarding the interval halving optimization method. The most significant drawback of the method is caused by the interval halving approach itself. Bisecting the time interval and fixing its boundaries adds constraints to the optimization problem that are not present in a real situation. Consequently, the global optimum of the problem cannot be reached with the method, even if MILP problems can be solved to global optimality. Due to the nature of the interval halving approach, it is also impossible to include the investment cost in the objective function. Hence, it is somewhat inconvenient to use the method as an optimization-based design method for energy storage systems with a large number of design variables.

Further algorithm development could increase the application area of the interval halving optimization method. Other MILP solvers and perhaps even mixed integer non-linear programming (MINLP) solvers could be applied in the model in the subproblem optimization. Thereby, the method could be able to solve a larger variety of optimization problems.

Apart from the restrictions of the optimization model, the mathematical assumptions in the optimization problem formulation also affect the reliability of the case study results. Some mathematical assumptions are made in order to attune the problem to fit the optimization approach, while some

assumptions are just made due to the lack of available RSOC operation data. The RSOC operation, for example, is assumed to be linear, even though the RSOC operation is slightly non-linear according to several sources [21,30]. The performance of the RSOC is, in practice, also highly dependent on the operation temperature [30], which is not taken into account in this study, since it would considerably complicate the problem formulation. Uncertain technical factors such as the time required to switch between operational modes and balance of plant energy losses etc. are not taken into account in the model either. It is, however, complicated to create a proper mathematical RSOC model, since ROSC technology is still in an early development phase. A more thorough optimization of the RSOCHS system would therefore have to be postponed until the technology is mature enough to provide sufficient technical and operational data.

The results of the optimization show that the operating cost benefits as well as the self-sufficiency of an RSOCHS system increase when more solar PV panels are installed. The RSOC and hydrogen storage system enables the consumer to utilize more of the generated PV power, which means that less energy has to be imported. An RSOCHS system could hence be an important contributor in achieving a net zero annual energy balance for individual buildings. In building energy systems with big solar PV panel installations, the annual imported energy could be halved by installing an RSOCHS system. This reduction in imported energy would mostly be in the form of a reduction in district heating.

However, installing an RSOCHS system does have a relatively low impact on the OPEX of the energy systems. The maximum capital savings in terms of annual OPEX is only about 5000 EUR, which is only a fraction of the investment cost of the hydrogen gas compressor [15]. The payback time of the RSOCHS system might thus be greater than the total life of the investment.

The optimal hydrogen storage size varies between 50 and 5000 kWh, depending on the RSOC size and the solar PV panel area. The optimal size of the hydrogen storage tends to increase as the RSOC size and the solar PV panel area increases.

The analysis of the ROSCHS system operation optimization showed that the hydrogen storage should be filled during the summer using both electricity generated by the solar PV and electricity imported from the grid. This behavior can be motivated by high PV generation during the summer period and the high district heating prices in the winter period. Because of the high district heating prices in the winter, it is economically feasible to charge the energy storage with electricity from the grid and use it for heating.

The case study in this paper only focuses on the optimal operation of the RSOCHS system and its operational benefits compared to a system without energy storage. To better understand the economic value of the RSOCHS system, a life-cycle cost analysis is required. RSOC technology is still in a development phase and the cost of an ROSCHS system is thus not yet competitive compared with other energy storage technologies [31]. Mass manufacturing of RSOC components is envisaged to bring down the investment cost of the technology, but exactly by how much is still difficult to predict. Hence, RSOC technology might play a significant role in future energy storage and power-to-gas systems.

**Author Contributions:** Conceptualization, O.L., R.W., F.P., A.H. and J.S.; methodology, O.L.; validation, O.L.; formal analysis, O.L.; investigation, O.L.; writing—original draft preparation, O.L.; writing—review and editing, O.L., A.H., R.W. and F.P.; visualization, O.L.; supervision, A.H., R.W. and F.P. All authors have read and agreed to the published version of the manuscript.

**Funding:** The authors would like to thank Business Finland. The work was part of Smart Otaniemi innovation ecosystem (Smart Otaniemi Pilot Phase 2, 8194/31/2018) financed by Business Finland.

**Conflicts of Interest:** The authors declare no conflict of interest.

## Nomenclature

| | |
|---|---|
| $a$ | The beginning of the year |
| $b$ | The end of the year |
| $C_{Pex,i}$ | Electricity export price at time step $i$ |
| $C_{Pim,i}$ | Electricity import price at time step $i$ |
| $C_{Qex,i}$ | Heat export price at time step $i$ |
| $C_{Qim,i}$ | Heat import price at time step $i$ |
| CPU | Central processing unit |
| $E_{H,0}$ | Stored hydrogen gas at the beginning and end of the year |
| $E_{H,cap}$ | Hydrogen gas storage capacity |
| $h_{c,in}$ | Specific enthalpy of hydrogen gas at the inlet of the hydrogen compressor |
| $h_{c,out}$ | Specific enthalpy of hydrogen gas at the outlet of the hydrogen compressor |
| LHV | Lower heating value of hydrogen gas |
| $m_{i,j}$ | Storage load at central point of time interval $j$ in optimization step $i$ |
| MILP | Mixed integer linear programming |
| $n$ | The number of the optimization step |
| $N$ | Number of time steps |
| OPEX | Operating expense |
| PV | Photovoltaic |
| $P_{d,i}$ | Electrical power demand of the building at time step $i$ |
| $P_{E,in,i}$ | Electrical power input to the RSOC at time step $i$ |
| $P_{E,in,max}$ | Maximum electrical input of the RSOC |
| $P_{E,in,min}$ | Minimum electrical input of the RSOC |
| $P_{ex,i}$ | Exported electrical power at time step $i$ |
| $P_{F,out,i}$ | Electrical power output of the RSOC at time step $i$ |
| $P_{F,out,max}$ | Maximum electrical output of the RSOC |
| $P_{F,out,min}$ | Minimum electrical output of the RSOC |
| $P_{im,i}$ | Imported electrical power at time step $i$ |
| $P_{PV,i}$ | Solar PV electrical power output at rime step $i$ |
| $\dot{Q}_{b,i}$ | Heat generated by combustion of hydrogen gas at time step $i$ |
| $\dot{Q}_{d,i}$ | Heating demand of the building at time step $i$ |
| $\dot{Q}_{ex,i}$ | Exported heat rate at time step $i$ |
| $\dot{Q}_{im,i}$ | Imported heat rate at time step $i$ |
| RSOC | Reversible solid oxide cell |
| RSOCHS | Reversible solid oxide cell and hydrogen gas storage |
| SA | Simulated annealing |
| SOEC | Solid oxide electrolysis cell |
| SOFC | Solid oxide fuel cell |
| $t$ | Time step size |
| $\delta_i$ | binary operation mode decision variable for the time step $i$ |
| $\eta_c$ | Efficiency of the hydrogen gas compressor |
| $\eta_{is}$ | Isentropic efficiency of the hydrogen gas compressor |
| $\eta_m$ | Electrical motor efficiency of the hydrogen gas compressor |

## Appendix A

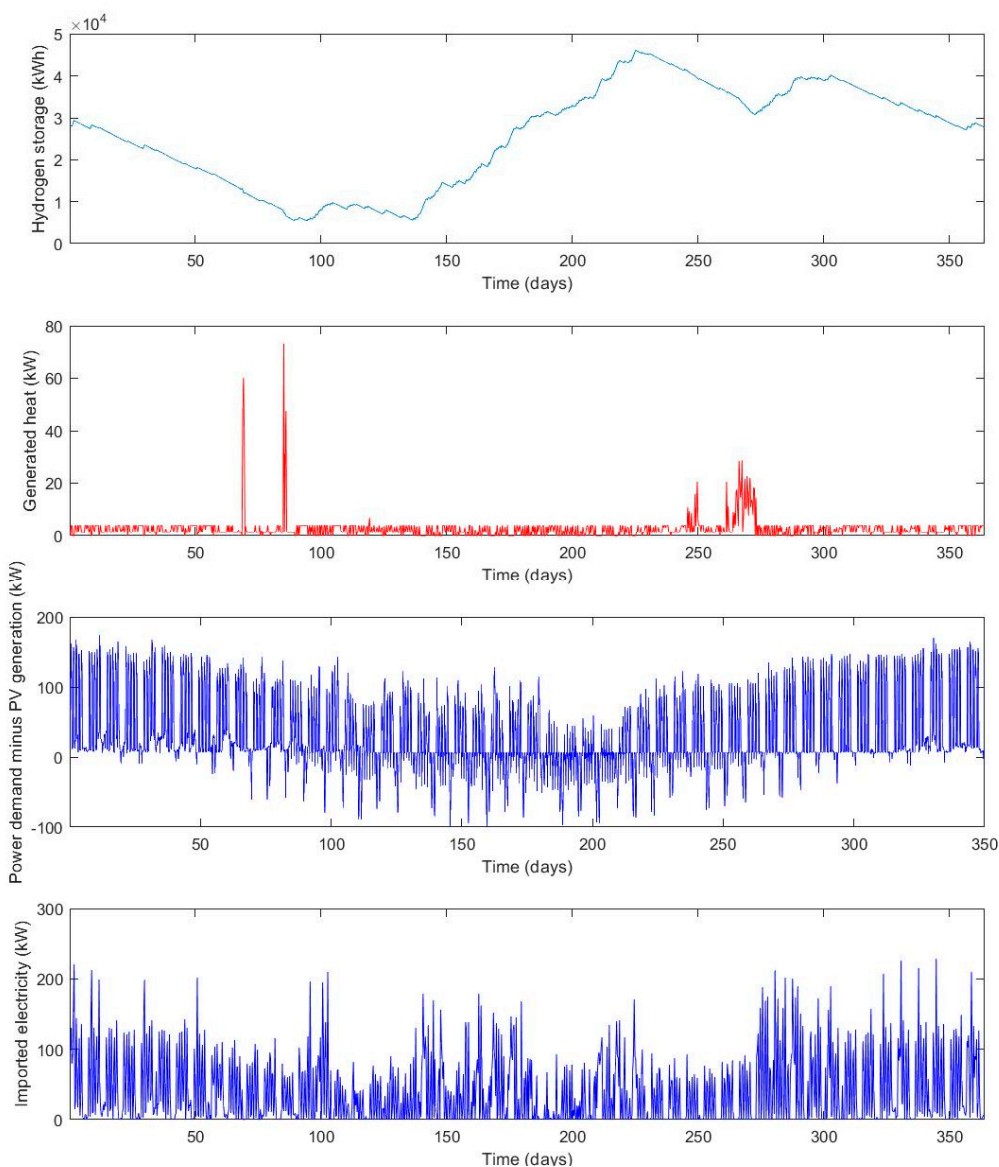

**Figure A1.** Optimal operation of a 20/80 kW RSOC system with 1000 m² solar PV panel.

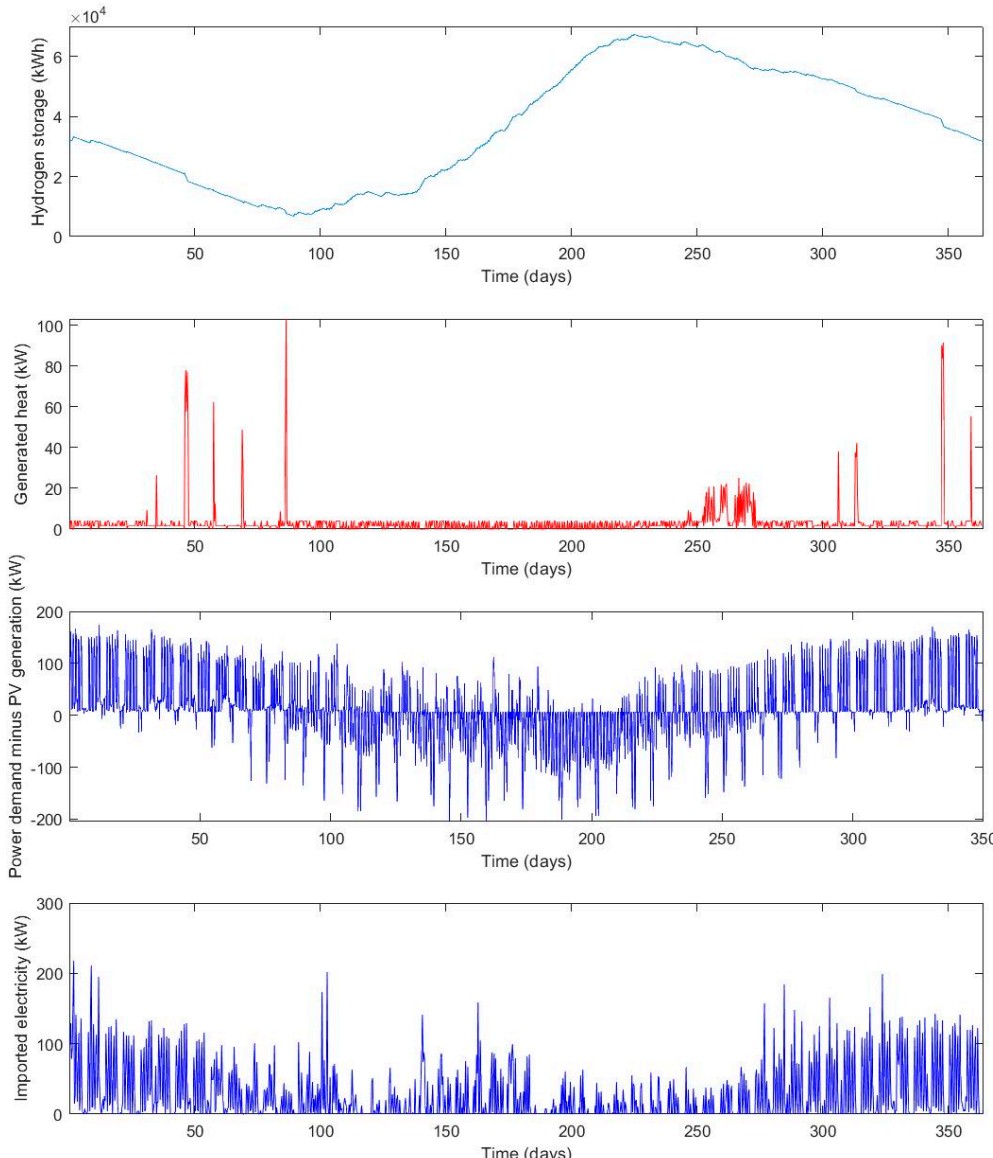

**Figure A2.** Optimal operation of a 20/80 kW RSOC system with 2000 m$^2$ solar PV panel.

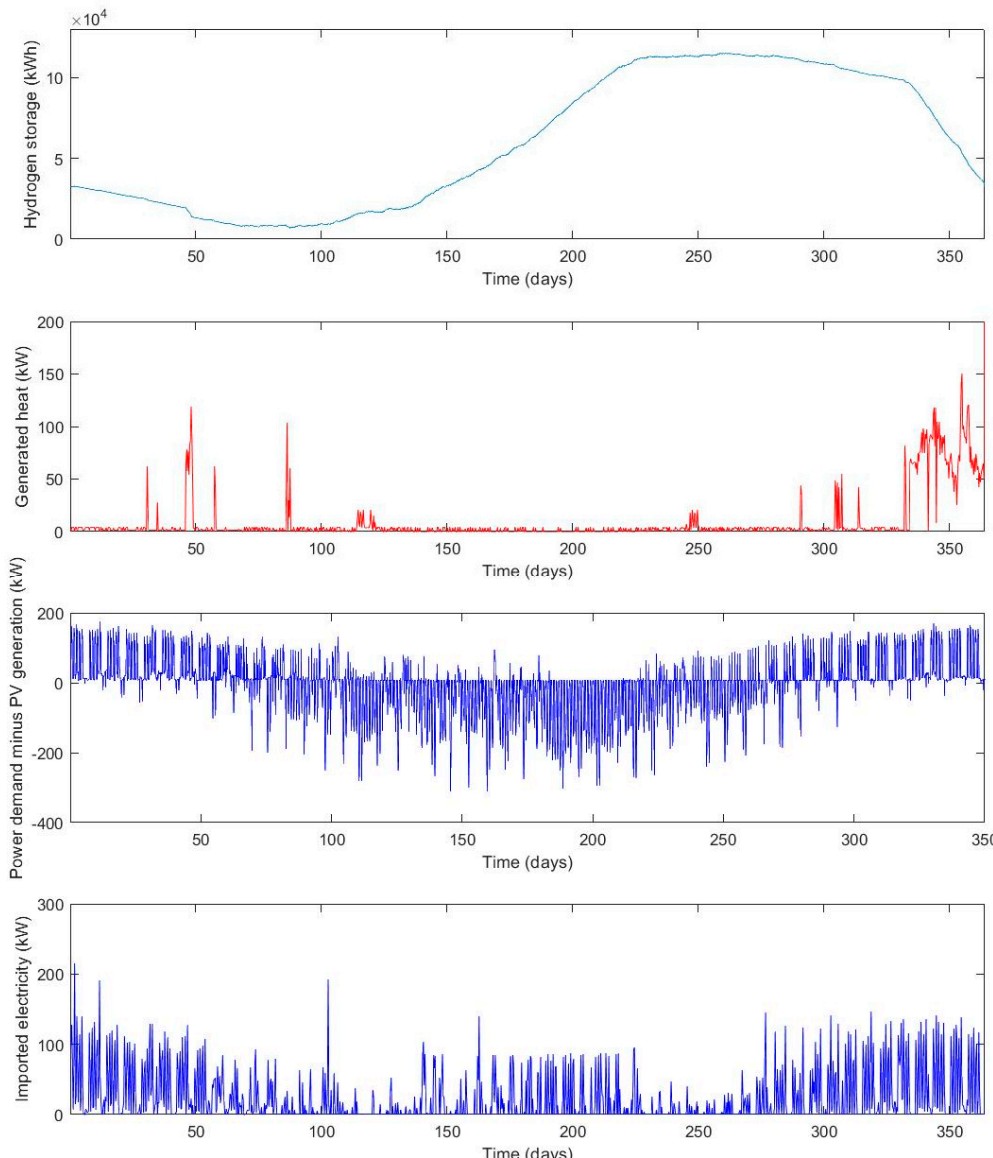

**Figure A3.** Optimal operation of a 20/80 kW RSOC system with 3000 m² solar PV panel.

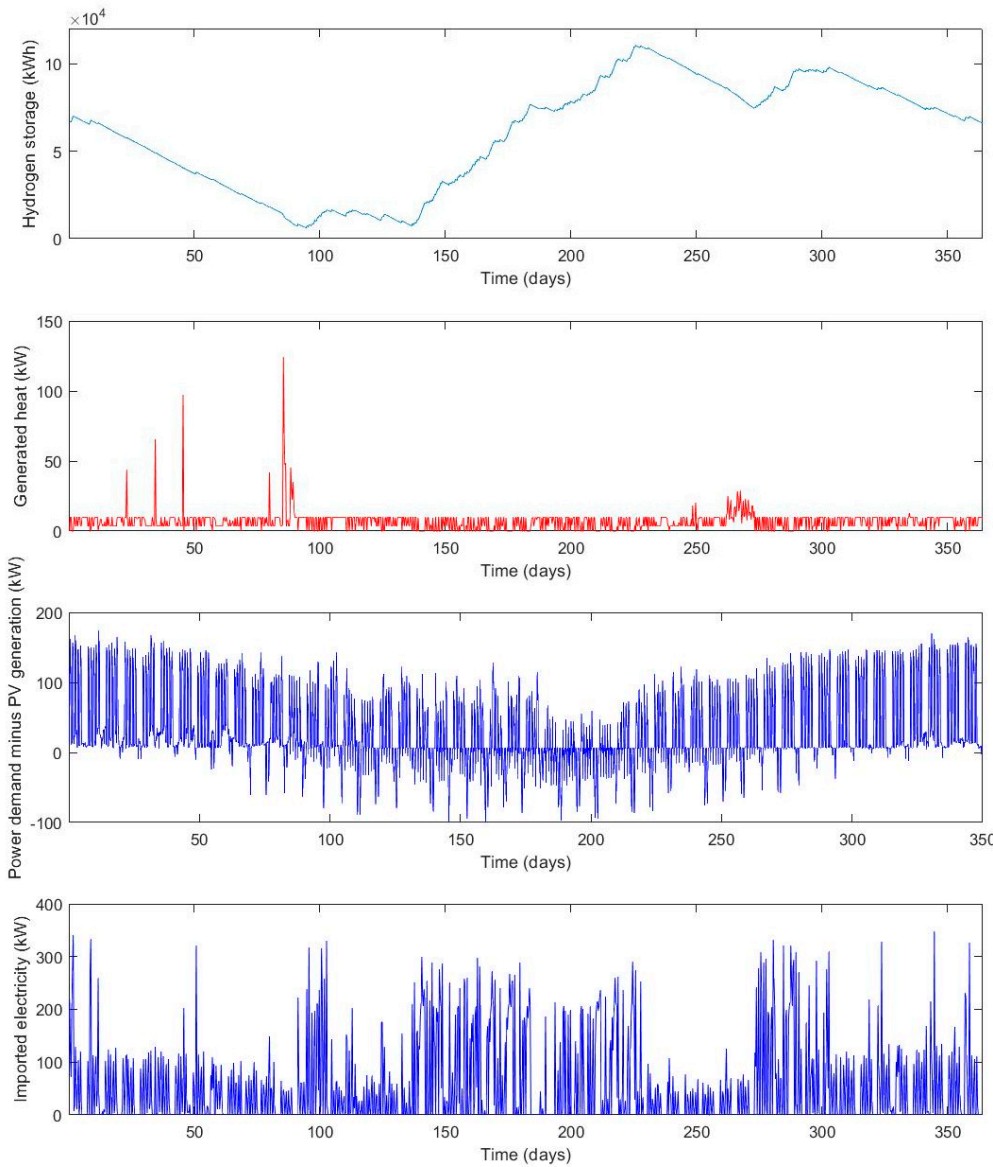

**Figure A4.** Optimal operation of a 50/200 kW RSOC system with 1000 m² solar PV panel.

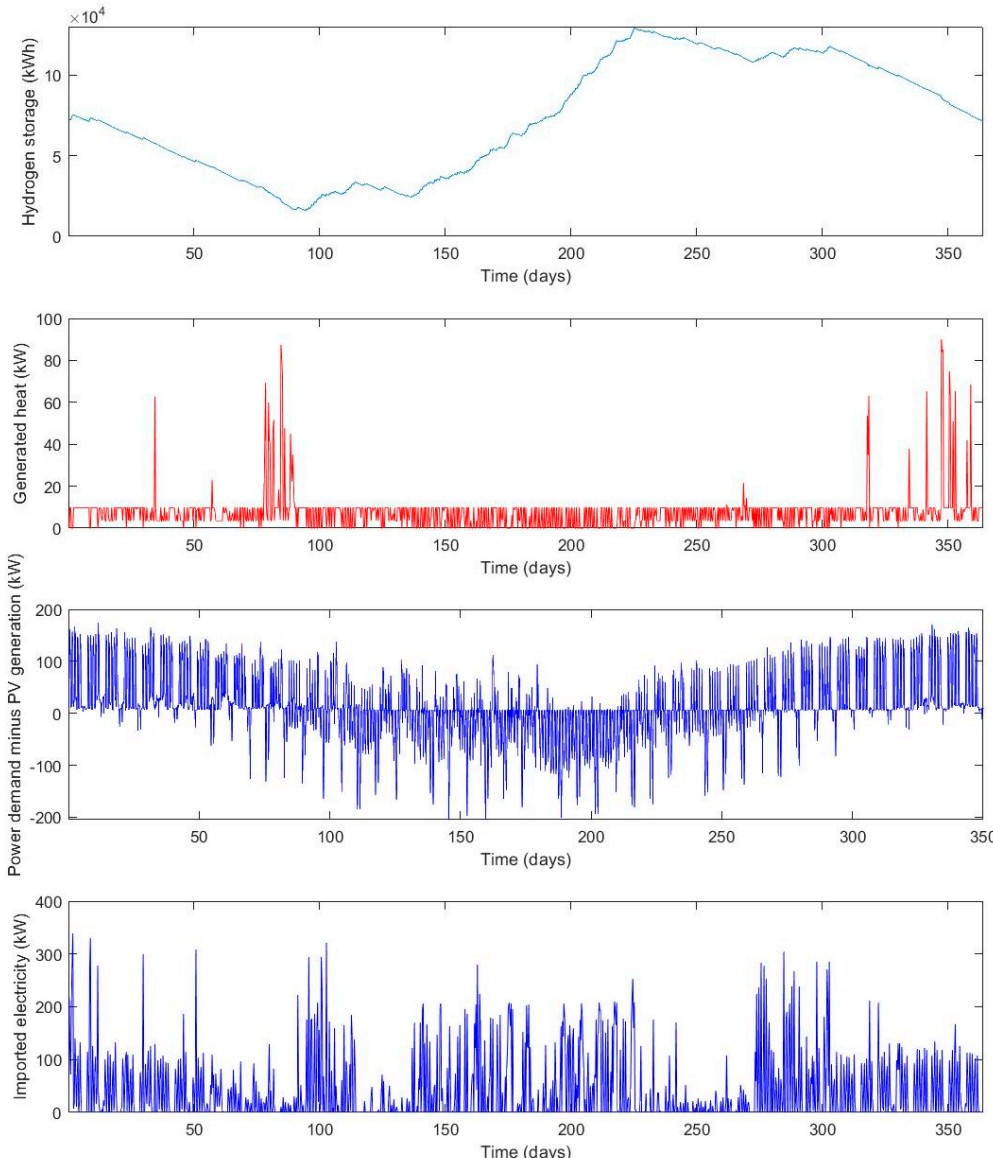

**Figure A5.** Optimal operation of a 50/200 kW RSOC system with 2000 m$^2$ solar PV panel.

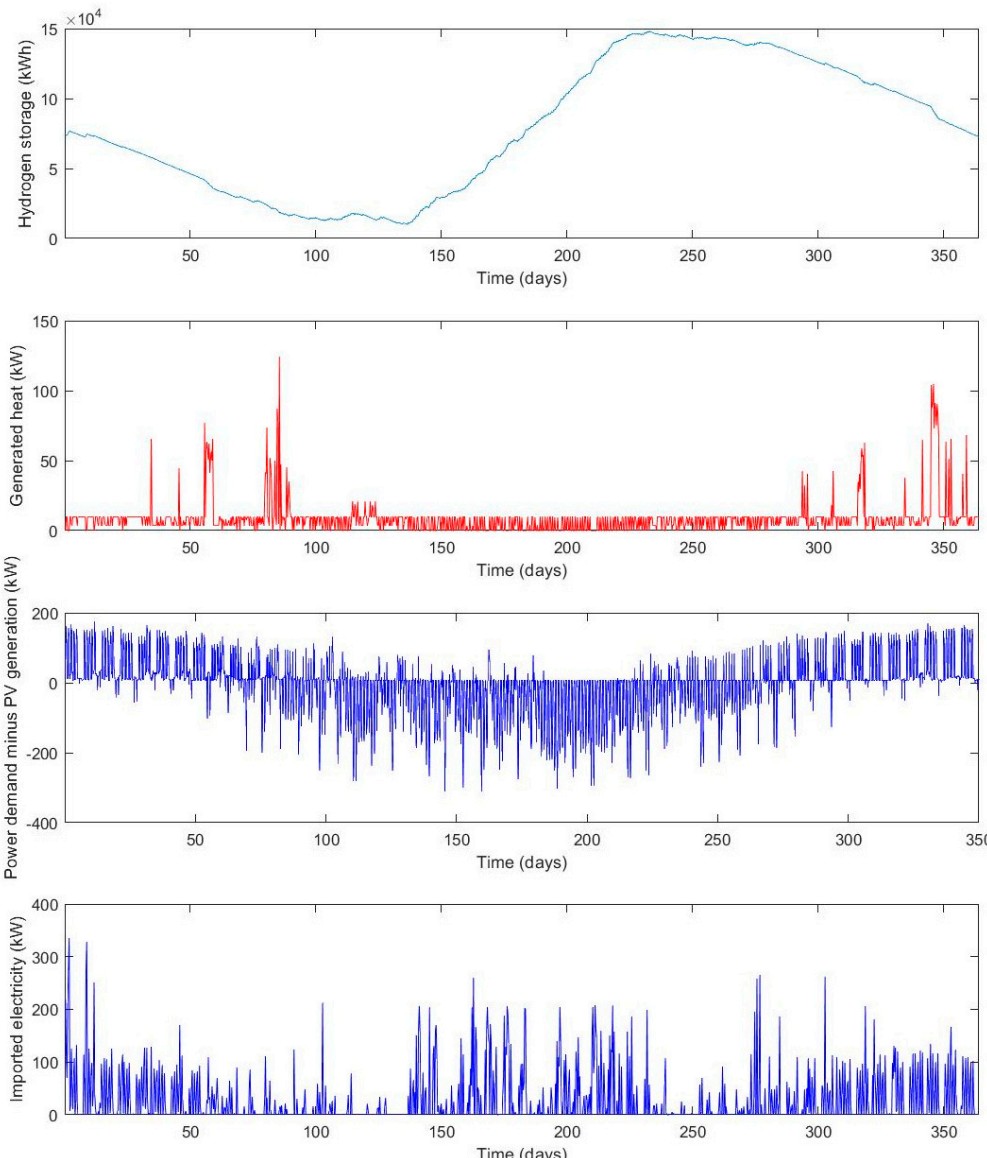

**Figure A6.** Optimal operation of a 50/200 kW RSOC system with 3000 m$^2$ solar PV panel.

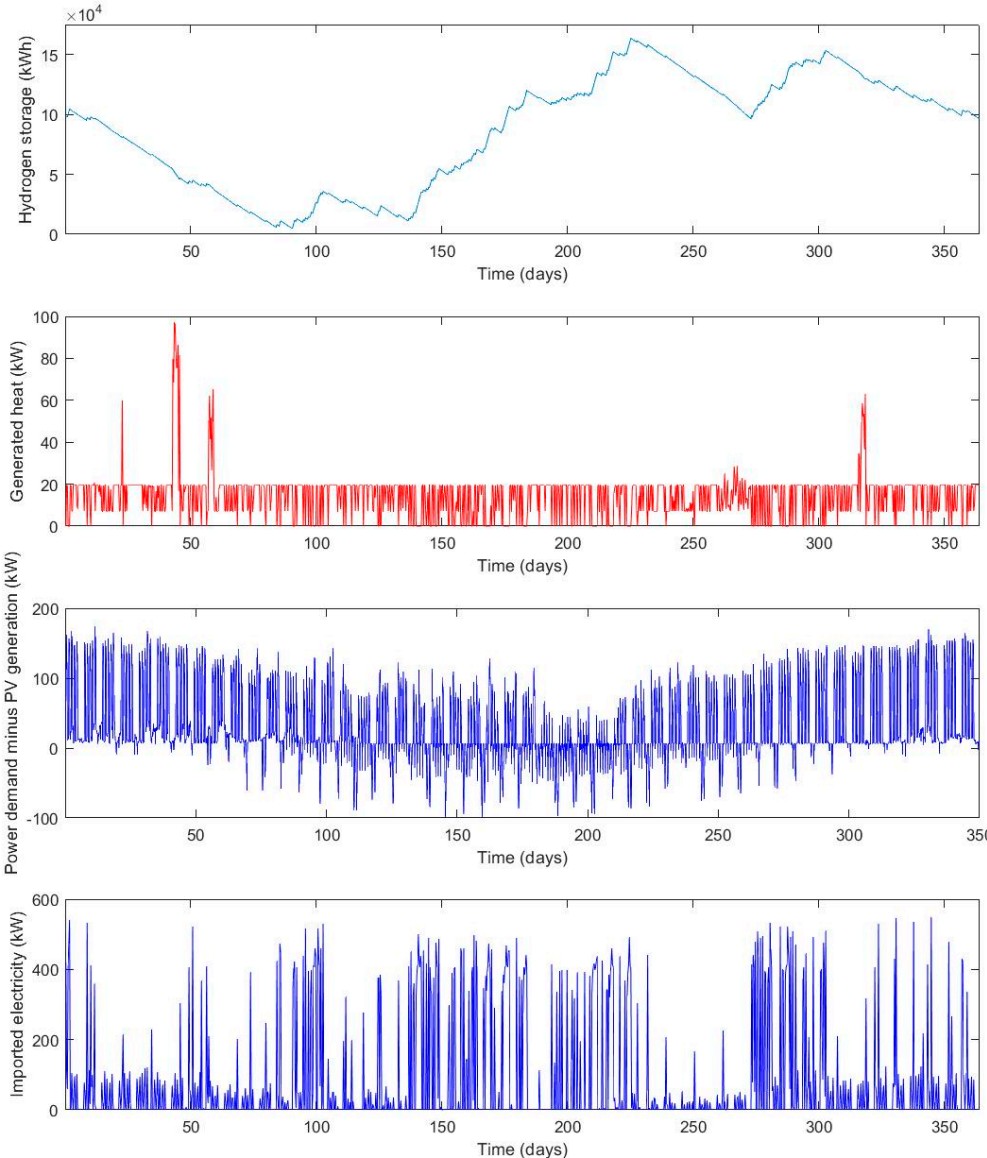

**Figure A7.** Optimal operation of a 100/400 kW RSOC system with 1000 m² solar PV panel.

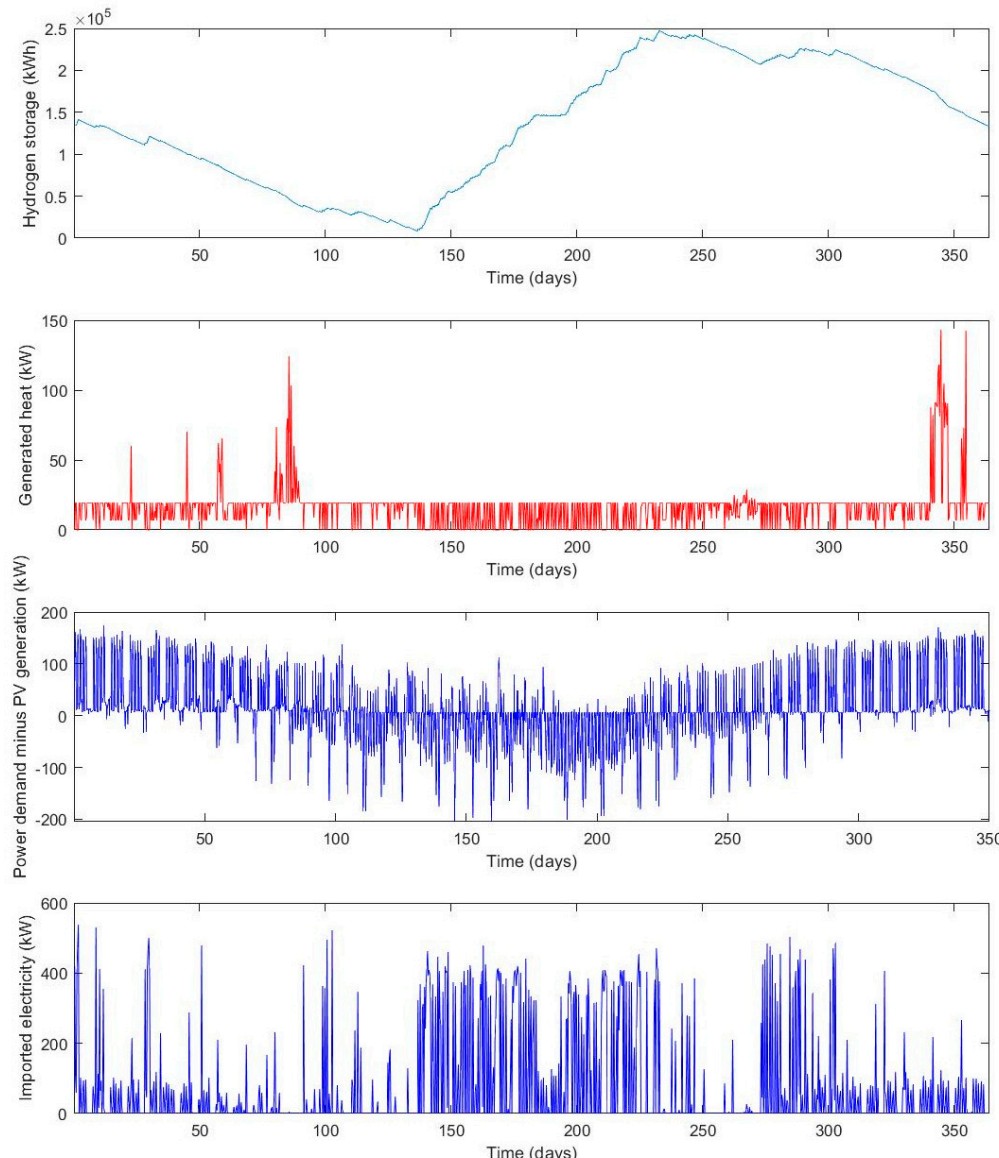

**Figure A8.** Optimal operation of a 100/400 kW RSOC system with 2000 m$^2$ solar PV panel.

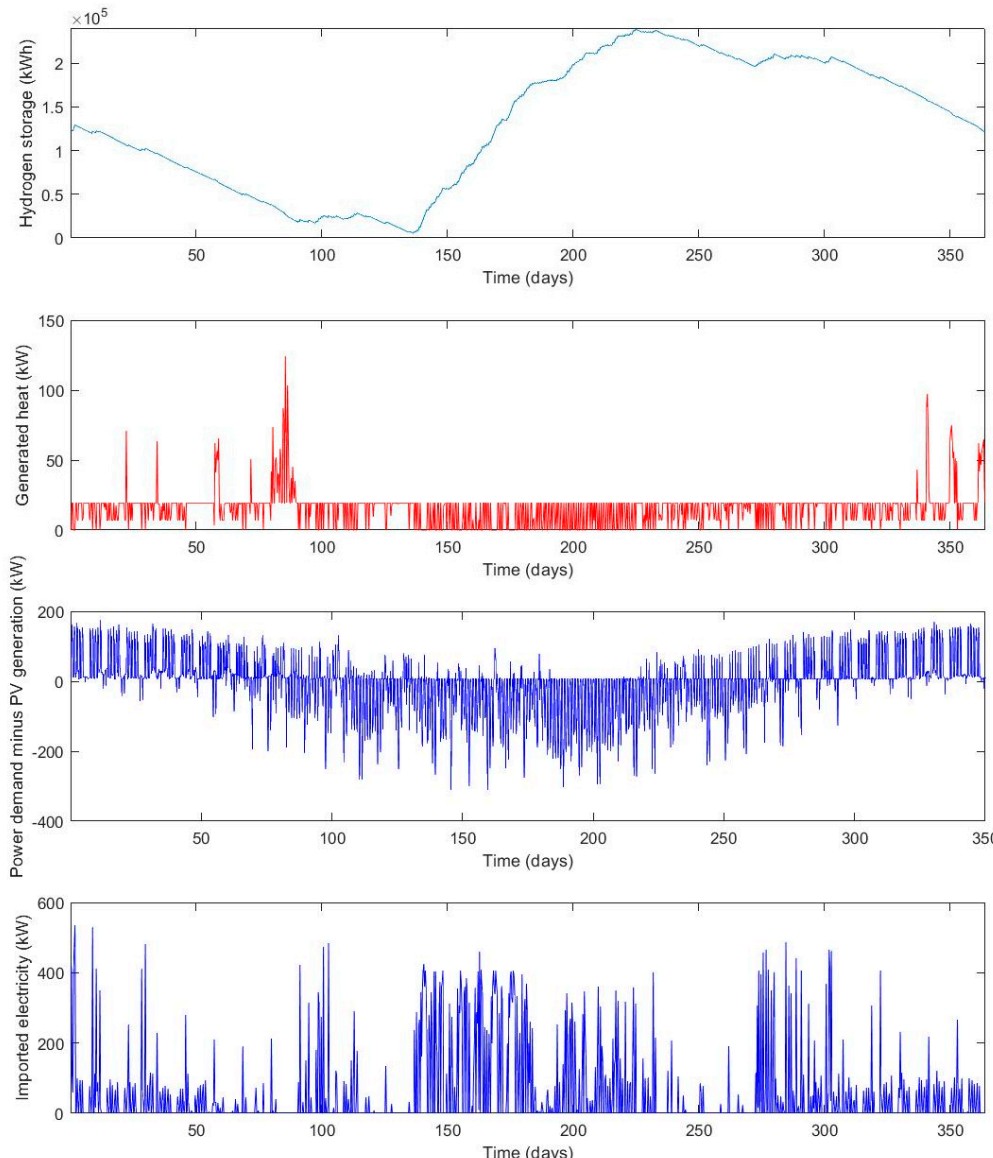

**Figure A9.** Optimal operation of a 100/400 kW RSOC system with 3000 m$^2$ solar PV panel.

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
