# Peer review of "A MILP Optimization Method for Building Seasonal Energy Storage: A Case Study for a Reversible Solid Oxide Cell and Hydrogen Storage System"

_buildings, doi:10.3390/buildings10070123_

Round 1
Reviewer 1 Report
The authors proposed a MILP solution algorithm based on solving successive problems over decreasing size partitions of the total horizon time, and where some of the decision variables of the child optimization problems are fixed from the results of their corresponding parent optimization problems. To show the performance of the MILP optimization method, the authors solve a case study where the operation of reversible solid oxide cell and hydrogen storage system is optimized.
In my view, this is an interesting contribution that proves the usefulness of design engineering systems by the tools provided by mathematical programming, in particular those that allow to model discrete decisions. The paper is clearly written with detailed explanations. Another strength of the paper is that the authors applied the model to a real case study. Before final acceptance the authors could address the following points:
The authors justify the development a optimization method because they tackle a problem that is computationally expensive, but to be more precise it would be useful to know the degree of such complexity by adding the model statistics (number of continuous variables, number of binary variables, number of constraints, CPU time for solving the problem, etc.). Depending on this assessment of the model complexity, it could occur that using the state-of-the-art MILP solvers instead of the Matlab solver “intlinprog”, the case study proposed by the authors can be solved in a unique step. In this case, a better solution than using the “interval halving optimization method” will be obtained as the latter is a sequential approach. In this regard, the authors claim in the conclusions section that their case study “is computationally too expensive to be solved by conventional optimization methods and that no other alternative optimization method suitable for the problem has been found in the literature so far”. In addition, they also state the requirement of assume liner behavior in their model to fit into the linear optimization approach followed. However, and specially, in the area of Process Systems Engineering, there has been significant advances in the development of MINLP solvers, which, perhaps, they could be successfully applied to optimize large-scale building design problems. Some of them can be found in these references (of course is not an exhaustive list, just to mention a few):
- Computers and Chemical Engineering. Volume 28, Issue 8, 15 July 2004, Pages 1169-1192
- Computers and Chemical Engineering. Volume 30, Issue 4, 15 February 2006, Pages 614-634
- Journal of Global OptimizationVolume 59, Issue 2-3, July 2014, Pages 503-526
In the results section, the authors show several figures where the performance of the system is a function of the solar PV area (for different RSOCHS systems). It seems that the higher the PV area the higher the system performance (for instance the OPEX decrease with an increasing PV area). Why did not the authors consider the investment cost of the solar PV panels (i.e. a CAPEX contribution) and combine in the objective function with the OPEX? This could affect the optimal results. In addition, if the Life Cycle Analysis is considered, the environmental impact of the solar PV panels used could have a significant contribution.
Author Response
Thank you for the great and thorough review!
A table with relevant optimization problem statistics has now been added to the manuscript.
State-of-the-art MILP solvers would without doubt be more accurate for small MILP otimization problems than the method presented in the paper. However, a sequential optimization approach will significantly reduce the size of the problem and might therefore be faster for problems with a large number of integer variables.
You mentioned that there have been significant advances in the development of MINLP solvers and that these solvers could be applied to large-scale building optimization problems. Combining the presented interval halving method with such solvers could be an interesting topic for further research. It would be interesting if such solvers could be able to solve similar problems as presented in the case study without significantly increasing the calculation time. We added a comment about this in the conclusions section.
About the CAPEX integration in the model; due to the characteristics of the interval halving method, adding the investment cost of the solar PV and hydrogen storage to the objective function is unfortunately not possible (at the moment). As the method is splitting the problem into several subproblems that are optimized separately, it is not possible to include variables that should have the same value in each subproblem (this is now clarified in the manuscript). The inability to include the CAPEX in the objective function is one of the greatest disadvantages of this optimization method.
Taking into account a life-cycle assessment in the optimization would be a great idea for future research within the area of net-zero and positive energy buildings and districts. In this study, however, we wanted to use a relatively simple cost optimization problem in order to analyze the performance of the presented method.
Kind regards,
Reviewer 2 Report
- The abstract should be more concise, and I suggest authors provide the background, target, significance, methodology, main results, and so on, in this abstract.
- Need strong comment on scientific outcomes.
Author Response
Thanks for the review!
1. The abstract has been modified in order to make it more concise, and the authors have tried to maximize content of the abstract, while still obeying the word count limit.
2. Scientific outcomes have now been addressed in the conclusions section.
Kind regards,
Reviewer 3 Report
Thank you for submitting your paper “A MILP optimization method for building seasonal energy storage: A case study for a reversible solid oxide cell and hydrogen storage system” to the Journal of Buildings.
The study is very interesting and overall it is well written but needs some improvements.
- The title of the paper should show the "intent or novelty”, in my opinion, it results too generic.
- I suggest reorganizing the abstract, highlighting the novelties introduced, it should contain answers to the following questions:
- What problem was studied and why is it important?
- What methods were used?
- What are the important results?
- What conclusions can be drawn from the results?
- What is the novelty of the work and where does it go beyond previous efforts in the literature?
The originality of the paper needs to be stated clearly.
- In the introduction, it could be useful to add a separate section relative to the stat of the arts, underlining the real innovation of the study. There are several recent studies on this issue:
- A novel energy-economic-environmental multi-criteria decision-making in the optimization of a hybrid renewable system, Sustainable Cities and Society, Volume 52, 2020, 101780, ISSN 2210-6707, https://doi.org/10.1016/j.scs.2019.101780.
- Hypothesis of thermal and mechanical energy storage with unconventional methods, Energy Conversion and Management, Volume 218, 2020, 113014, ISSN 0196-8904, https://doi.org/10.1016/j.enconman.2020.113014.
- Energy and economic optimization of solar-assisted heat pump systems with storage technologies for heating and cooling in residential buildings, Renewable Energy, Volume 157, 2020, Pages 90-99, ISSN 0960-1481, https://doi.org/10.1016/j.renene.2020.04.121.
- The potential for integration of hydrogen for complete energy self-sufficiency in residential buildings with photovoltaic and battery storage systems, International Journal of Hydrogen Energy, 2020, ISSN 0360-3199, https://doi.org/10.1016/j.ijhydene.2020.04.170.
- In the methodology, it is important to put in evidence the following points:
- The proposed methodology has numerous advantages, such as:
- The main novelties and objectives of the present work can be summarized as:
- All the graphs should be explained much more.
- It might be useful to identify the location with a global climate classification, such as the “Köppen classification”.
- Encyclopædia Britannica, Encyclopædia Britannica, inc., 2020, https://www.britannica.com/science/Koppen-climate-classification
- Using the Köppen classification to quantify climate variation and change: An example for 1901–2010. Environmental Development, 6, 69-79, 2013, 10.1016/j.envdev.2013.03.007.
- The conclusions should be more focused on results, including the main numerical values.
- Please, add the table of the nomenclature.
Author Response
Thank you for the thorough review!
- The title of the paper has not been changed as the authors consider it describing enough.
-
The abstract has been modified in order to give a clearer picture of the novelty of the paper. Additionally, the authors have tried to maximize content of the abstract, while still obeying the word count limit.
- A couple of extra references was added to the litterature review. These references are expanding the background in optimization of energy storage systems for buildings with integrated PV panels.
- The fist paragraph in the methodology section was modified so that it gives a clearer picture of what the novelty of the method is.
- The authors thought that one paragraph per graph provides enough information. Thus, no further explanations were added.
- The location of the case study building is now defined with coordinates and the climate of the location is defined by the Köppen climate classification.
- Key numerical values and further observations from the result section is now presented in the conclusions section.
- A table of nomenclature has been added to the manusctipt.
Kind regards,